

# Expanding the simulation of East Asian Super Dust Storm: Physical transport mechanism impacting the Western Pacific

Steven Soon-Kai Kong [1], Saginela Ravindra Babu [1], Sheng-Hsiang Wang [1], Stephen M. Griffith [2], Jackson Hian-Wui Chang [1, 3], Ming-Tung Chuang [4], Guey-Rong Sheu [1, 5,*], Neng-Huei Lin [1, 5,*]

[1] Department of Atmospheric Sciences, National Central University, Taoyuan, 32001, Taiwan
[2] Department of Atmospheric Sciences, National Taiwan University, Taipei, 10617, Taiwan
[3] Preparatory Center for Science and Technology, University Malaysia Sabah, Jalan UMS, 88400, Kota Kinabalu, Sabah, Malaysia
[4] Research Center for Environmental Changes, Academia Sinica, Taipei, 11529, Taiwan
[5] Center for Environmental Monitoring and Technology, National Central University, Taoyuan, 32001, Taiwan

*Correspondence to*: Neng-Huei Lin (nhlin@cc.ncu.edu.tw) and Guey-Rong Sheu (grsheu@atm.ncu.edu.tw)

**Abstract.** Dust models are widely applied over the East Asian region for the simulation of dust emission, transport and deposition. However, due to the uncertainties in estimates of dust flux, these methods still lack the necessary precision to capture the complexity of transboundary dust events. This study demonstrates an improvement in the Community Multiscale Air Quality (CMAQ) model dust treatment during long-range transport of dust from northwest China to the South China Sea (SCS). To accomplish this, we considered a super dust storm (SDS) event in March 2010, and evaluated the dust scheme by including adjustments to the recent calibration (Dust_Refined_1) and bulk density (Dust_Refined_2) refinements individually and in combination (Dust_Refined_3). The Dust_Refined_3 normalized mean bias of $PM_{10}$ was -30.73 % for the 2010 SDS event, which was lower compared to Dust_Refined_1 (-41.34 %) and Dust_Refined_2 (-50.09 %). Indeed, the Dust_Refined_3 improved the simulated AOD value during significant dust cases, for instance, in March 2005, March 2006 and April 2009. Dust_Refined_3 also showed more clearly that in March 2010, a 'double plume' (i.e., one plume originated from the Taiwan Strait and the other from the Western Pacific) separated by the Central Mountain Range (CMR) of Taiwan Island affected dust transport on Dongsha Island in the SCS. On 15-21 April 2021, both CMAQ simulations and satellite data highlighted the influence of typhoon 'Surigae' on dust transport to downwind Taiwan and the Western Pacific Ocean (WPO). The CMAQ





Dust_Refined_3 simulations further revealed a large fraction of dust aerosols were removed over WPO due to typhoon 'Surigae'. Hence, the model indicated near-zero dust particle concentration over the WPO, which was significantly different from previous dust transport episodes over the Taiwan region. Therefore, our study suggested an effective method to improve dust management of CMAQ under unique topographical and meteorological conditions.

## 1 Introduction

Dust storms are a major source of dust aerosols and particles in outdoor air pollution, with significant health, environmental and ecological impacts adjacent to and downwind of dust source regions, especially in the East Asian region (Shao and Dong, 2006; Griffin and Kellogg, 2004; Yao et al., 2021). Likewise, dust aerosols can significantly affect the Earth's climate through direct and indirect influences on the radiation balance of the atmosphere (Chen et al., 2017b; Huang et al., 2014; Dong et al., 2019). The Gobi Desert (GD) in northern China and Mongolia, and the Taklamakan Desert (TD) in western China are dust storm hotspot regions in East Asia. Several studies have reported on the impacts of this East Asian Dust (EAD), particularly the effects during springtime on air quality and air pollution over source regions (e.g. northern China) and over downwind regions such as Korea, Japan, and Taiwan (Bian et al., 2011; Han et al., 2012; Guo et al., 2017; Jing et al., 2017; Dong et al., 2016; Jiang et al., 2018; Kong et al., 2021, 2022; Tan et al., 2017; Uno et al., 2017). Fugitive dust can be dispersed over thousands of miles; thus, regional and large-scale meteorological conditions play a crucial role in the transport of these dust particles during dust storms.

A series of dust storms (15 March; 27 March; and 15 April) occurred over the Gobi Desert area in the spring of 2021 including one of the largest dust storms in the past decade (15 March; "3.15" dust storm hereafter). This severe dust storm turned the sky into sepia over Beijing (Sullivan, 2021), with maximum $PM_{10}$ concentrations reaching up to 7,400 μg m$^{-3}$. A few studies investigated the origin, transport processes, and impact of the "3.15" dust storm on air quality by multi-source observations and numerical modeling (Liang et al., 2022; Gui et al., 2022; Jin et al., 2022; He et al., 2022). Gui et al. (2022) reported the detailed spatial, temporal, and vertical evolution of the "3.15" dust storm and 27



March ("3.27" dust storm) events by utilizing satellite dust optical depths, lidar dust extinction profiles, visibility measurements, and RGB Himawari imagery. Further, Jin et al. (2022), described the dust sources, aerosol optical, microphysical, and radiative properties, and meteorological drivers of the three events in 2021 spring. Even though past studies of more mild dust storm events have shown impacts as far afield as the Taiwan region (Kong et al., 2021, 2022), most of the studies regarding 2021 super dust storm (SDS) events were focused on the impact and transport over China and eastern downwind parts of Asia. None of the studies reported the transport of dust from these events to the South China Sea (SCS), including Taiwan, and also chemical-transport modeling of these events was limited.

On the other hand, several numerical modeling studies have been conducted to simulate March 2010 SDS event (Bian et al., 2011; He et al., 2022; Li et al., 2011; Zhao et al., 2011; Lin et al., 2012a; Park et al., 2012; Chow et al., 2014; Chen et al., 2017a). Fortuitously, this SDS event was also detected (Wang et al., 2011; 2012) over Dongsha Island (i.e. Pratas Island, 20°42052" N, 116°43051" E) in the northern SCS during the Dongsha Experiment (http://aerosol.atm.ncu.edu.tw), which as part of the 7-SEAS (the Seven South East Asian Studies; http://7-seas.gsfc.nasa.gov, Lin, et al., 2013) project was designed to investigate the weather-aerosol interaction over Southeast Asia. Although the SDS arrival at Dongsha Island was only described based on ground measuring and satellite imagery (Wang et al., 2011; 2012), these studies showed the possibility of transporting dust aerosol from northwest China to the SCS boundary layer. However, a detailed high-resolution numerical modeling system is needed to clarify the movement of the SDS aerosols in the region.

In previous our studies (Kong et al., 2021, 2022), we simulated moderate-intensity dust events at the surface and at higher altitudes over the Taiwan region by using the Weather Research and Forecasting-Community Multiscale Air Quality (WRF-CMAQ) model. Recognizing the opportunity to model SDS events impacting Taiwan and the SCS, in this study we utilized the WRF-CMAQ model with the latest windblown dust treatment to characterize the transport mechanism of the SDS events over these downwind regions. As the notable amount of atmospheric mineral received by SCS over the past years, that influences the oceanic ecosystem, a more detailed investigation regarding long-range transport of dust episodes over the region can be vital (Duce et al., 1991; Wang et al., 2012). The





present manuscript is organized as follows. The methodology of the WRF-CMAQ model setup and dust
treatment calibration are discussed in Section 2. The results and discussion are presented in Section 3.
Finally, the summary and conclusions obtained from the present study are summarized in Section 4.
**2. Data and Methodology**
**2.1 WRF-CMAQ model setup and dust treatment calibration**
CMAQ is a state-of-the-art air quality model developed by the United States Environmental Protection
Agency USEPA (Appel et al., 2013) that distinguishes 19 chemical species within the dust particles,
thus providing a detailed description of dust mineralogy (Dong et al., 2016). Heterogenous chemistry
between the gas and aerosol phase also occurs (e.g. mechanisms) and can affect the dust chemical
composition, thus the gas-phase module is also activated in the model. This work utilized WRFv3.9.1
for the meteorological field prediction, and CMAQ v5.3.3 to simulate the transport of SDS on 18-24
March 2010, and several well-known severe dust storms, for instance, on 17-19 March 2005, 18-20
March 2006, 25-27 April 2009 and 23-21 April 2021 (Wang et al., 2012; Jin et al., 2022). The modeling
domain was set up to cover East Asia (d01), including the Gobi Desert, with a resolution of 81 km and
nested towards Taiwan at a resolution of 27 km (d02), 9 km (d03a), and 3 km (d04a) (Fig. 1a). The
nesting of Dongsha Island with 9 km and 3 km resolution (d03b and d04b) was set up to specifically
capture the long-range transport over the SCS. The model consisted of 40 vertical layers, with 8 layers
below ~1 km altitude, 13 layers below ~3 km altitude, and 27 layers covering the upper layer to ~21 km.
The initial and lateral boundary conditions of the model were constructed using the NCEP FNL re-
analysis dataset on a 1° ×1° grid. The data assimilation was conducted by grid-nudging in all domains.
The CB06 gas-phase chemical mechanism and AERO7 aerosol module model were implemented in
CMAQ for the present study.

105       Anthropogenic emission inventories in East Asia were obtained from the MICS-Asia (Model

Inter-Comparison Study for Asia) Phase III emission inventory (Li et al., 2017). Biogenic emissions for
Taiwan were prepared by the Biogenic Emission Inventory System version 3.09 (BEIS3, Vukovich and
Pierce, 1988), and for regions outside Taiwan by Model of Emissions of Gases and Aerosols from



Nature v2.1 (MEGAN, Guenther, et al., 2012). TEDS 9.0 (Taiwan Emission Database System, TWEPA,
2011; https://erdb.epa.gov.tw/) was used for domain 4 (d04a) covering the Taiwan region, for the years
2005, 2006, 2009 and 2010, and TEDS 10.0 (TWEPA, 2021; https://erdb.epa.gov.tw/) was used for the
year 2021. Since domain d04b was specifically downscaled to Dongsha Island, no anthropogenic
emissions were applied for the region.
Five simulation scenarios including Dust_Off, Dust_Default, Dust_Refined_1, Dust_Refined_2,
and Dust_Refined_3 are presented and described in Table 1. The inline dust treatment was not included
in Dust_Off. For Dust_Default, wind speed, soil texture, and surface roughness length were integrated
based on the scheme by Foroutan et al. (2017). The performance of Dust_Off and Dust_Default in
simulating a moderate dust episode was compared by Kong et al. (2021), but this comparison has not
been investigated for a Super Dust Storm. This comparison provides important information as CMAQ is
often run for air quality purposes but with only Dust_Off or Dust_Default; yet, dust influence in that
observation data would be underestimated if using these basic schemes, thus reporting this performance
could be useful to later studies. The latest dust treatment over East Asia proposed by Kong et al. (2021)
was implemented in the Dust_Refined_1 scenario, which reduced the soil moisture at the surface and
revised the source-dependent species profile. The bulk soil density ($\rho b$) should be revised to represent
the real soil type in China, which is represented by Dust_Refined_2 (Liu et al., 2021). As the default
bulk soil density ($\rho b$) is set to 1,000 kg m$^{-3}$ in CMAQ for all soil types, the soil condition in China is not
specifically represented in the Dust_Default and Refined_1 scenario. Hence, the $\rho b$ of sand, loam,
sandy clay loam, and clay were revised as 1,550, 1,350, 1,450, and 1,300 kg m$^{-3}$, respectively, for
Dust_Refined_2 (Yu et al., 2015; Liu et al., 2021). Finally, Dust_Refined_3 combined the
Dust_Refined_1 and Dust_Refined_2 schemes.
**2.2 Measurements at the downwind sites**
The Dongsha Experiment included multiple platforms of instruments such as the
NASA/GSFC/COMMIT (Chemical, Optical, and Microphysical Measurements of In-situ Troposphere;
http://smartlabs.gsfc.nasa.gov) mobile observatory, the Taiwan Environmental Protection
Administration (TEPA) mobile facility, and a lidar system (EZ-Lidar; Leosphere Co.), of which detailed



information can be found in the literature (Wang et al., 2011). Briefly, continuous $PM_{10}$ and $PM_{2.5}$ mass
concentrations were measured by a Tapered Element Oscillating Microbalance (TEOM; Model 1400 ab;
R&P Co.), which draws in air to a sample filter and changes the oscillation frequency of a calibrated
tapered element. This change in frequency is then converted to a particle mass based on the restoring
force constant of the tapered element. Moreover, a VAISALA WXT520 meteorological sensor was
specifically set up at Dongsha for the field campaign. It was used to measure weather conditions near
the surface, such as horizontal wind speed, wind direction, and precipitation. The dataset from Dongsha
Experiment was used to validate the CMAQ model precision during the dust storm event in March 2010.
In addition, the hourly $PM_{10}$ concentration datasets from the Cape Fuguei, Wanli, Pingzhen, Hsinchu,
Xitun, Xinying, Zuoying, and Daliao sampling sites in Taiwan were obtained from the website of the
Taiwan Environmental Protection Agency (https://data.epa.gov.tw/).
**2.3 Reanalysis products and satellite measurements**
The Modern Era Retrospective-analysis for Research and Application version 2 (MERRA-2, Gelaro et
al., 2017) reanalysis data were used in this study to demonstrate the spatiotemporal distribution of dust
and compare it with the air quality model, irrespective of the cloud cover. MERRA-2 is a NASA
reanalysis ($0.5° \times 0.625°$ resolution) utilizing Goddard Earth Observing System Data Assimilation
System Version 5 (GEOS-5) and assimilates remotely sensed data. Besides, the level-3 MODIS AOD at
550 nm (MYD08) were used. The daily MODIS data was obtained from the AQUA platform with $1° \times$
$1°$ resolution. Apart from this, we also used daily mean merged precipitation data from the Global
Precipitation Mission (GPM) satellite in the present study. MERRA-2 data can be accessed through the
NASA Goddard Earth Sciences Data Information 135 Services Center (GES DISC;
https://disc.gsfc.nasa.gov/), while MODIS and GPM datasets were downloaded from the GIOVANNI
official website (https://giovanni.gsfc.nasa.gov/giovanni/).





## 3 Results and Discussion

### 3.1 CMAQ model evaluation

The statistical analysis of the CMAQ $PM_{10}$ modeling performance for the March 2010 SDS event is shown in Table 2. DUST_Off and DUST_Default were similarly underestimated (Normalized Mean Bias (NMB) = -65.02 % and -54.34 %, respectively), compared with the observed values, which is consistent with the results of Dong et al. (2016) and Kong et al. (2021) that simulated moderate-intensity dust events. The Dust_Refined_1 and Dust_Refined_2 simulations exhibited improved accuracy (NMB = -41.43 % and -50.09 %, respectively), highlighting the importance of revising the dust treatment before simulating the SDS event over a downwind region (Kong et al., 2021). Moreover, the NMB for Refined_1 was lower than Refined_2 suggesting that simply calibrating the bulk soil density is not as effective as calibrating for soil moisture fraction and dust emission speciation. Eventually, Dust_Refined_3 resulted in the best performance (NMB = -30.73 %). Our results indicate the importance of including both calibration methods in order to reduce the model uncertainty.

Figure 2 shows the in-situ and CMAQ-simulated $PM_{10}$ concentrations at Wanli station (representing a background location in northern Taiwan) and Dongsha Island (representing the northern South China Sea region) during 19-24 March 2010. In both locations, the Dust_Off trend vastly underestimated the observations, whereas Dust_Default showed increased $PM_{10}$ concentrations but still resulted in an underestimation. The $PM_{10}$ concentration at Wanli reached 1000 μg m$^3$, which is the maximum range of the instrument. CMAQ model predicted a peak $PM_{10}$ concentration of 868.8 μg m$^3$, thus was 13.1 % lower than the observation result, but should be considered a conservative estimate as the observation amount may be underrepresented. At Dongsha Island, Dust_Refined_1 generated a higher peak $PM_{10}$ value (371.6 μg m$^3$) compared to Dust_Refined_2 (255.3 μg m$^3$). Likewise, Dust_Refined_3 generated a peak concentration of 524.4 μg m$^3$, the highest among all of the simulation scenarios, and only 5.9 % lower than the maximum observed $PM_{10}$ concentration of 557.0 μg m$^3$.

Daily average modeled $PM_{10}$ concentration differences between Dust_Off and other simulations over the East Asia region during 19-23 March 2010 is shown in Fig. 3. Dust_Default showed $PM_{10}$



concentration differences of approximately 200 μg m$^{-3}$ over the source region of northwest China.
Dust_Refined_1 exhibited a difference of ~ 600 μg m$^{-3}$ over the source region, which was greater than
Dust_Refined_2. Overall, Dust_Refined_3 produced > 800 μg m$^{-3}$ difference, which was the highest
among the simulations. This result was further verified over the downwind region, where high PM$_{10}$
concentrations were observed in Taiwan and SCS regions (Fig. 3h). Further, we plotted MERRA-2
surface dust concentrations during 20-21 March 2010, which are shown in Fig. S1. The MERRA-2 data
indicated the dust plume only impacted Taiwan, while did not arrive at the SCS. Our model, on the
other hand, clearly (apparently) simulated the arrival of the dust plume to Dongsha Island, which is
consistent with 7-SEAS Dongsha Experiment-measured PM$_{10}$. Hence, this effort emphasizes the
importance of utilizing high-resolution simulations for depicting dust pollutant transport episodes.
Besides that, the wind speed and wind direction at different elevation levels play an important role in
dust transport. Uncertainty could be from this data if the wind component is poorly captured by the
satellite images used to generate the MERRA-2 reanalysis data.
In order to re-emphasize the precision of the dust treatment, we then implemented our
calibration method for other dust storm episodes that transported dust from northern Taiwan toward
southern Taiwan, which were documented by Wang et al. (2012). Hence, we carried out the 3-day
averaged sensitivity test over the East Asia region, estimated from d01 for four other notable dust storm
cases: 17-19 March 2005, 18-20 March 2006, 25-27 April 2009 and 20-22 March 2010 (Table 3).
Generally, DUST_Refined_3 performed well in simulating AOD over the East Asia region throughout
the four strong dust storm events. The average AOD value of the DUST_Refined_3 yielded an NMB of
-16.02 %, which was markedly better than DUST_OFF (-26.09 %), DUST_Default (-25.24 %),
DUST_Refined_1 (-19.58 %) and DUST_Refined_2 (-24.40 %). Improvement of the modeled AOD by
approximately 10 % was comparable with the result suggested by Dong et al. (2016). The temporal and
spatial distribution of CMAQ AOD showed the DUST_Refined_3 can modestly capture the dust storm
pattern as compared to MODIS-daily average AOD (Fig. S2). These results suggested DUST_refined_3
should be used for calibration as it successfully uplifts the dust aerosol at the source region and
simulates the notable dust cases over the East Asia region.



## 3.2 Role of Central Mountain Range (CMR) on dust transport

Figure 4 shows the spatial distribution of CMAQ estimated $PM_{10}$ concentrations under Dust_Refined_3 simulations over East Asia during the March 2010 event. A low-pressure system of approximately 996 hPa over northwest China was associated with the uplifting of dust. As shown in Fig. 4b, a strong pressure gradient led to strong wind speed generation, thus pushing the dust aerosol to move in the southeast direction (Song et al., 2019; Kong et al., 2022). The dust arrived at massive concentrations in transboundary regions such as southern China, Japan, Korea, and Taiwan, consistent with previous studies (Lin et al., 2012; Bian et al., 2012) (Fig. 4c).

Figure 5 shows the CMAQ $PM_{10}$ spatial distribution under Dust_Refined_3 simulations, depicting the dust transport over Taiwan and Dongsha Island. On 16 UTC 20 March, one dust cloud reached the surface in the Taiwan region (Fig. 5a) and split into two particular dust plumes due to the Central Mountain Range (CMR) located in the center of Taiwan (Fig. 5b). At 04 UTC 21 March, the first dust plume arrived at Dongsha Island, followed by the second 4 hours later (Fig. 5d, e). The model result suggested the separated dust plumes originated from two different directions: the first one from the Taiwan Strait (P1) and the second one from the Western Pacific Ocean (P2a) (Fig. 2). Meanwhile, the measured $PM_{10}$ concentration at Dongsha Island showed two peak values, at 15 UTC 21 March and 04 UTC 22 March 2010, respectively. The trends of the observed Dongsha peak value were consistent with the CMAQ model results, where the model exhibited a clear $PM_{10}$ peak at 06 UTC March 2010 (P2b in Fig. 2b). The "tail" of the dust plume swept over the South China Sea including Dongsha Island due to the easterlies and northeasterly wind (Red arrow in Fig. 5e). Then, the dust cloud gradually dissipated, leaving Dongsha Island and moving to southern China.

To better understand the role of the CMR on the SDS transport over SCS and Dongsha Island, we carried out another simulation by removing the CMR and setting a zero altitude for the whole of Taiwan Island within the WRF. We then examined the vertical profiles of the $PM_{10}$ simulation, by categorizing the model depiction into Cross A, Cross B, Cross C, and Cross D (Fig. 1b). The multiple cross-section lines indicated the vertical dust pattern at different stages or locations, such as the dust arrival at East China Sea (Cross A), Central Taiwan (Cross B), and the front (Cross C) and backward





(Cross D) of Dongsha Island across South China Sea. At 18 UTC on 20 March, preceding arrival to
Taiwan, both simulations with and without CMR showed the same pattern of PM$_{10}$ over the East China
Sea (ECS) (Fig. 6a, 6b). At 00 UTC on 21 March, the CMR of Taiwan effectively separated the dust
cloud into two parts as shown in the control run (Fig. 6c), which is not seen in the simulation without
CMR (Fig. 6d). Due to the role of the CMR, CMAQ simulations indicated two dust plumes arriving to
Dongsha Island (Cross C, Fig. 6e). Meanwhile, only one single plume was presented by the simulation
without CMR (Fig. 6f). At 15 UTC 21 March, both dust plumes were merged together and transported
to the west and northwest directions with respect to the easterly wind (Fig. 6g).
The role of CMR has been discussed in the literature, as it alters the strength of frontal
systems as they pass by Taiwan (Chien and Kuo, 2006). Also, due to the channel effect between the Wu
Yi Mountains in southeastern China and CMR in Taiwan, the air flow is forced to accelerate and causes
high intensity wind speeds through the Taiwan Strait (Lin et al., 2012a). Thus, the differential wind
speeds over the Taiwan Strait and eastern Taiwan, owing to the CMR, apparently caused uneven
"double plumes" over the Taiwan region.

## 3.3 Role of the meteorological condition on dust transport

The observed PM$_{10}$ over Dongsha Island (Fig. 2b) shows two separate peaks on March 20 and 22,
consistent with the reports of Wang et al. (2011). Our observed data showed minimal PM$_{10}$
concentrations between the two peaks, even though no precipitation was recorded over the site (Fig S3).
Figure 8 shows the daily precipitation over the downwind region. As discussed in Section 3.2, abundant
dust aerosol was transported through the Taiwan Strait and the Western Pacific Ocean, before arriving
at Dongsha Island. During 19-20 March 2010, no rainfall was captured by the satellite data over both
marine regions, resulting in the high PM$_{10}$ concentration of the first peak (Fig. 7a, b). On the other hand,
from 21 March to 22 March of 2010, heavy rainfall occurred in eastern Taiwan around the Western
Pacific Ocean. (Fig 7c, d). Based on the Global Precipitation Mission (GPM) satellite dataset,
precipitation in the region may have washed away dust aerosols before reaching the SCS and Dongsha
Island, resulting in lower PM$_{10}$ concentrations.





Regarding the importance of precipitation and wet deposition during the dust transport over the
downwind areas (Li et al., 2011; Kong et al., 2021), the spatial distribution of the modeled wet
deposition is shown in Fig. 8. Obviously, wet deposition was more intense over ECS than SCS, with
~20 mg m$^{-2}$ and ~6 mg m$^{-2}$, respectively. However, in Fig. 2, the modeled PM$_{10}$ concentration over
Wanli (northern Taiwan) was more underestimated than that at Dongsha Island (SCS). This situation
may be related to differences in the wet deposition magnitude over the different marine boundary layers.
Revising the CMAQ model deposition mechanism over the marine boundary layer was vital as
highlighted in our previous study (Kong et al., 2021). In the present work, we again suggest the
possibility of deposition flux variability over a different part of the marine boundary region (ECS vs.
SCS), which has not been mentioned by Kong et al. (2021).

## 3.4 Role of a Typhoon on a dust storm event in April 2021

Several studies have discussed the multiple dust storms over China in the spring of 2021 and the
associated dust emissions, transport/deposition, and radiative impact (Jin et al., 2022; Gui et al., 2022;
He et al., 2022; Liang et al., 2022; Tan et al., 2022). However, these studies only analyzed the incident
over the continental region. The SDS in transboundary areas, especially across the ocean marine
boundary layer, has not been closely tracked. As shown in Fig. 9(a), in the year 2021, three intensive
dust storms occurred during 14-18 March, 27-30 March and 15-17 April over China, which contained
the primary dust source region in each event (https://www.aqistudy.cn/). In the cities of northern China,
including Beijing, Hohhot and Taiyuan, the observed hourly PM$_{10}$ concentrations vastly exceeded 1000
µg m$^{-3}$. Figure 9(b) shows the PM$_{10}$ and PM$_{2.5}$ time series over Cape Fuguei (a background site in
northern Taiwan) during the spring of 2021 (https://data.epa.gov.tw/). Three PM$_{10}$ peaks of of 165 µg
m$^{-3}$, 116 µg m$^{-3}$, and 246 µg m$^{-3}$, were observed at 07 UTC 17 March, 13 UTC 22 March, and 22 UTC
18 April 2021, respectively. According to the Hybrid Single-Particle Lagrangian Integrated Trajectory
model (HYSPLIT) backward trajectory, the dust plumes arriving on 22 March and 18 April originated
from the Gobi Desert (Figure S4). The event on 17 March was from southern Japan, passed through the
marine boundary layer, and may have been due to local dust pollution from the local beach area. In
other words, out of three significant East Asian dust storms, one reached Taiwan and caused air quality





degradation over the region. The sudden increase in $PM_{10}$ mass concentration that exceeded 200 μg m$^{-3}$
indicated the high possibility of an SDS (Song et al., 2022).
Figure S5 shows the spatial distribution of AOD at 550 nm over East Asia from MERRA-2
reanalysis data and CMAQ Dust_Refined_3 simulations. Generally, the model reproduced well the dust
transport pattern shown by MERRA-2 reanalysis data during the dust event on 18 April.  Figure 10
shows the spatial distribution of surface dust mass concentrations over East Asia during 18-19 April
2021. At 00 UTC on 18 April, the dust cloud arrived in Taiwan and approached the SCS. Meanwhile,
Typhoon Surigae, located east of the Philippines, accelerated and pulled a significant amount (up to 50
μg m$^{-3}$) of dust toward and into the typhoon center (Fig. 10b-e). Eventually, the dust mass
concentrations around the typhoon reduced (Fig. 10f-10h), while another fraction of the dust plume
passed through Taiwan and the Taiwan Strait, and was further transported towards the SCS.
The influence of the typhoon system on the dust aerosol can be further quantified by comparing
the MERRA-2 hourly averaged dust mass concentration over the ECS, Western Pacific Ocean (WPO),
and SCS (Fig. 11). The difference between the maximum values and the mean averaged (11-25 April
2021) dust mass concentrations was the highest over the WPO (69.2 μg m$^{-3}$), compared with ECS and
SCS (13.6 μg m$^{-3}$ and 14.2 μg m$^{-3}$), indicating the remarkable dust removal by the typhoon. The
peripheral circulation on the southern side of the typhoon played a role in directing dust aerosol toward
the WPO and away from the ECS and SCS (Fig. 12a-d). This situation was due to the extreme wind
speed and the cyclonic rotation of the typhoon. The total precipitable water vapour shown by MERRA-
2 was intense around the eye of the typhoon, and the dust aerosol was shown to be washed out as it
passed through this area of the typhoon (Figure 12e-h). Moreover, the intensity of the total precipitation
was associated with the dust pattern (Li et al., 2011; Kong et al., 2021), as areas with more precipitation
(i.e. near the center) also contained lower dust concentrations (Fig. 10d-e).
As a result, the abnormal transport pattern can be attributed to the high-pressure system in
mainland China pushing the dust aerosol toward the downwind region (Chuang et al., 2008; Kong et al.,
2021), while the typhoon system over the Western Pacific Ocean accelerated transport of the dust plume



southward (Fig. 10i). CMAQ captured quite well the long-rang transport of dust toward the SCS and Dongsha Island, where the plume passed through the Taiwan Strait (10j-10l). However, no dust aerosol was found over the Western Pacific Ocean and the redirection of the dust plume by the typhoon, as illustrated by the MERRA-2 data, was not reproduced by the model.

Figure 13a-d shows the CMAQ daily dust wet deposition over East Asia, where a cluster of wet deposition was heavily distributed over the eastern Philippines. This large deposition flux could be related to the heavy rainfall from the typhoon (Fig. 13i-l). Also, a similar pattern was found for the dry deposition over the region, but with less intensity compared to the wet deposition (Fig. 13e-h). Nevertheless, the dry deposition was spread widely over the western Pacific Ocean, consistent with the daily mean wind speed over the region (Fig. 13m-p). Hence, the low dust concentration ($< 5$ μg m$^{-3}$) over the WPO as predicted by CMAQ may have been driven by dry deposition associated with the extreme wind speed triggered by the typhoon system.

Tropical cyclone (Typhoons/Hurricanes) normally occur over the WPO during the summer and fall seasons, and tend to impact air quality and enhance the rainfall over the region (Lin et al., 2011; Lam et al., 2018; Lin et al., 2021). Typhoons have been shown to increase aerosols over central Taiwan and create strong easterly flow causing stable weather conditions and weak wind speed, on the lee side of the CMR, i.e. in western Taiwan (Lin et al. 2021). The present study highlights the ability of a typhoon to remove dust aerosol that have been transported thousands of kilometers from northwest China. This enhanced wet deposition flux is consistent with Kong et al. (2021) that showed the influence of a rainfall belt to increase dust deposition over ECS.

The daily mean surface dust mass concentrations on 18 March 2005 (D1), 19 March 2006 (D2), 24 April 2009 (D3), 21 March 2010 (D4) and 18 April 2021 (D5) are displayed in Fig. 14. Episode D4 was a more intense dust plume compared to D1, D2, D3 and D5 as D4 was the SDS while the other episodes were just the regular dust storm (Wang et al., 2012; Wang et al., 2021). Episodes D1, D2, D3 and D4 revealed a common/typical dust transport pattern with the initial dust arrival at ECS, and then Taiwan Strait and WPO. However, in episode D5, the dust plume was only distributed over the ECS



and Taiwan Strait, and near-zero dust concentration was observed over the WPO. Hence, we revealed
an influence of a typhoon on dust transport patterns over East Asia, and highlighted the associated
excessive rainfall as an extraordinary, albeit irregular, removal mechanism over the WPO. As a result of
this variable transport pattern, the accuracy of the dust model in simulating the dust event encountered a
large degree of uncertainty, which is compounded by uncertainties in the dust emission scheme and dust
removal process (Kong et al., 2021; He et al., 2022). For instance, dust emission at the source region
can vary due to the different calibration methods, revealing the use of the dust scheme is not
straightforward and extensive testing should be carried out in order to achieve a better model
performance. Moreover, over the downwind region, the specific meteorological situation including the
wind speed, rainfall distribution, and extreme weather pattern could impact the transport pattern, and
further influence the dust model precision.

## 355 **4. Summary and Conclusions**

Dust storm outbreaks in East Asia are an irregular occurrence, but can rapidly deteriorate air quality
over a wide swath of the continent, causing severe health and environmental problems. Long-range
transport of East Asian dust to the South China Sea and the source emission, transport pattern and
deposition that facilitate these episodes have been largely overlooked. In this study, we combined
ground observations from the 7-SEAS Dongsha Experiment, MERRA-2 reanalysis, and MODIS
satellite images for evaluation and improvement of the CMAQ dust model for cases of EAD reaching
the Taiwan region, including Dongsha Island in the northern South China Sea.
We improved the dust treatment in the CMAQ model by implementing a refined aerosol profile,
the soil moisture fraction (Kong et al., 2021), and the bulk density of different soil types (Liu et al.,
2021). Based on the latest refined dust model, we simulated the long-range transport of a Super Dust
Storm (SDS) during 18-24 March 2010, and several significant dust storm events on 17-19 March 2005,
18-20 March 2006, 25-27 April 2009 and 15-21 April 2021, and detailed their respective transport
mechanisms. For the 2010 March SDS, our model suggested the dust simulation over Taiwan and
Dongsha Island was optimized with the dust scheme considering all the calibration methods, which is



the Dust_Refined_3 that provided the best NMB ( -30.73 %), compared to the calibration recommended
by Kong et al. (2021) (-41.43 %) and Liu et al. (2021) (-50.09 %). The SDS transport mechanism over
Dongsha Island in the South China Sea was influenced by the CMR in Taiwan. A "double plume" effect
was proposed, i.e. the dust plume split with a portion passing through the Taiwan Strait (west side of
CMR) and the other through the Western Pacific Ocean region (east side of CMR). Also,
Dust_Refined_3 treatment provided an optimized AOD simulation value during the significant dust
cases on March 2005, March 2006, April 2009 and March 2010.
In spring 2021, multiple East Asian dust storms occurred over the region after a period of
relative infrequency of nearly 12 years. One episode reached northern Taiwan, and deteriorated the
ambient air quality, resulting in a maximum $PM_{10}$ concentration of 246 μg m$^{-3}$. In contrast with previous
dust episodes that have reached the Taiwan region, both the satellite dataset and model result illustrated
a "double synoptic pattern" driven by a high-pressure system over the continent and a typhoon system
in the Western Pacific Ocean. The dust plume was pushed by the high-pressure system toward Taiwan,
and at the same time by typhoon "Surigae", resulting in the dust cloud splitting and a portion drawn in
by the typhoon circulation towards its center. This unique mechanism appeared to be accompanied by
increased dry or wet deposition of the dust particles over the WPO.

## Data Availability

MERRA-2 data are available online through the NASA Goddard Earth Sciences Data Information
Services Center (GES DISC; https://disc.gsfc.nasa.gov; last access: 08 June 2023). MODIS data used in
this study are available at https://asdc.larc.nasa.gov/(last access: 08 June 2023). The GPM dataset were
downloaded from the GIOVANNI official website at https://giovanni.gsfc.nasa.gov/giovanni/ (last
access: 08 June 2023). The observational data at Dongsha can be ordered by contacting corresponding
authors.



## Author Contribution

**Steven Soon-Kai Kong**: Conceptualization; Data curation; Formal analysis; Investigation; Methodology; Software; Validation; Visualization; Writing – original draft; Writing – review and editing.

**Saginela Ravindra Babu**: Conceptualization; Investigation; Methodology; Formal analysis; Writing – review and editing.

**Sheng-Hsiang Wang:** Formal analysis; Data curation.

**Stephen M. Griffith:** Writing – review and editing.

**Jackson Hian-Wui Chang:** Data curation and software.

**Ming-Tung Chuang:** Data curation.

**Guey-Rong Sheu:** Funding acquisition; Resources.

**Neng-Huei Lin**: Conceptualization; Visualization; Supervision; Funding acquisition; Resources; Writing – review and editing.

## Competing Interest

The authors declare that they have no conflict of interest.

## Acknowledgments

We acknowledged the Ministry of Science and Technology of Taiwan, under Project No. MSTC111-2811-M-008-069 for supporting the research. We also acknowledged the staff at Dongsha Island, and EPA Taiwan for the provision of the ground-based measurement datasets. We are also thankful to MERRA-2 and MODIS for the satellite product.

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





**Table 1** Summary of the design of the simulations used in the present study.

| Scenarios | Descriptions |
|---|---|
| Dust_Off | Without in-line calculation of dust. |
| Dust_Default | With the new default wind-blown dust treatment (Foroutan et al., 2017). |
| Dust_Refined_1 | Refined the soil moisture factor and the dust emission speciation profile for the Gobi Desert as suggested by Kong et al. (2021). |
| Dust_Refined_2 | Refined the bulb soil density according to China's soil type as suggested by Liu et al. (2021). |
| Dust_Refined_3 | Considering the both of Dust_Refined_1 and Dust_Refined_2. |

**Table 2** Statistical index for $PM_{10}$ concentration during 19-23 March 2010, for Taiwan Island (Banqiao,
Pinzhen, Hsinchu, Xitun, Xinying, Zhuoyin, Daliao) and Dongsha Island.

|  | Benchmark | Off | Default | Refined_1 | Refined_2 | Refined_3 |
|---|---|---|---|---|---|---|
| MeanObs |  | 176.47 | 176.47 | 176.47 | 176.47 | 176.47 |
| MeanMod |  | 50.99 | 64.71 | 82.12 | 70.60 | 96.23 |
| NMSE |  | 2.11 | 1.52 | 1.17 | 1.36 | 1.03 |
| MFB | ± 60% | -65.67 | -55.77 | -45.39 | -52.34 | -38.82 |
| NMB | ± 85% | -65.02 | -54.34 | -41.34 | -50.09 | -30.73 |
| NME | 85% | 65.02 | 60.35 | 57.44 | 59.16 | 55.24 |
| FAC2 | 0.5–2.0 | 0.70 | 0.83 | 0.98 | 0.87 | 1.11 |
| R | > 0.35 | 0.25 | 0.37 | 0.39 | 0.41 | 0.38 |

Note: the definition of the statistical formulas NMSE: Normalized Mean Square Error; MNB: Mean
Normalized Bias; NMB: Normalized Mean Bias; NME: Normalized Mean Error; FAC2: Factor of Two;
R: Correlation Coefficient.
**Table 3** CMAQ AOD evaluation against MODIS daily observation with Normalized Mean Bias (NMB)
for the multiple simulation scenarios during the dust storm episode of Mar2005 (16-20 March 2005),
Mar2006 (17-21 March 2006), Apr2009 (24-28 April 2009) and Mar2010 (19-23 March 2010).

| Cases | Mar2005 | Mar2006 | Apr2009 | Mar2010 | Mean |
|---|---|---|---|---|---|
| Dust_Off | -13.04 | -30.84 | -37.30 | -49.26 | -26.09 |
| Dust_Default | -13.04 | -30.84 | -37.30 | -45.03 | -25.24 |
| Dust_Refined_1 | -9.70 | -27.95 | -27.90 | -32.35 | -19.58 |
| Dust_Refined_2 | -13.04 | -30.84 | -37.30 | -40.80 | -24.40 |
| Dust_Refined_3 | -6.35 | -25.07 | -24.76 | -23.89 | -16.02 |




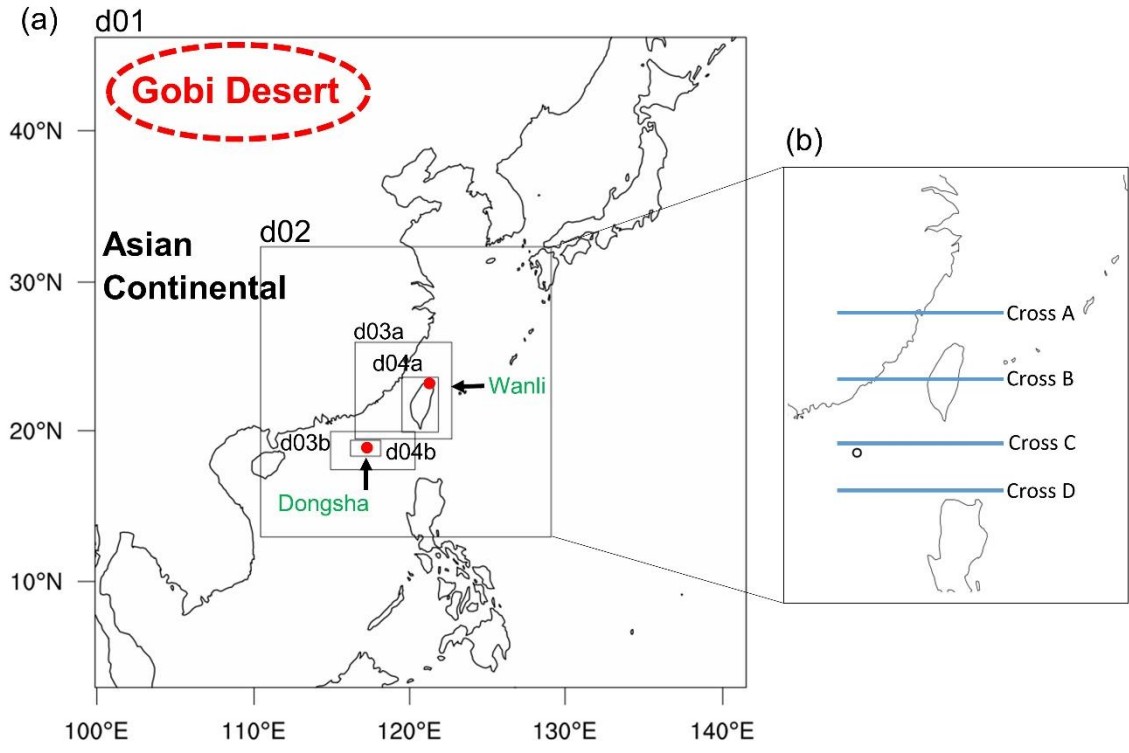

**Figure 1**: (a) Modeling domain configuration used in the present study. The red dot representing the location of the observation sites at Wanli and Dongsha. (b) The blue lines represent the transects that the dust plumes travelled along in this studies that are discussed in **Section 3**.





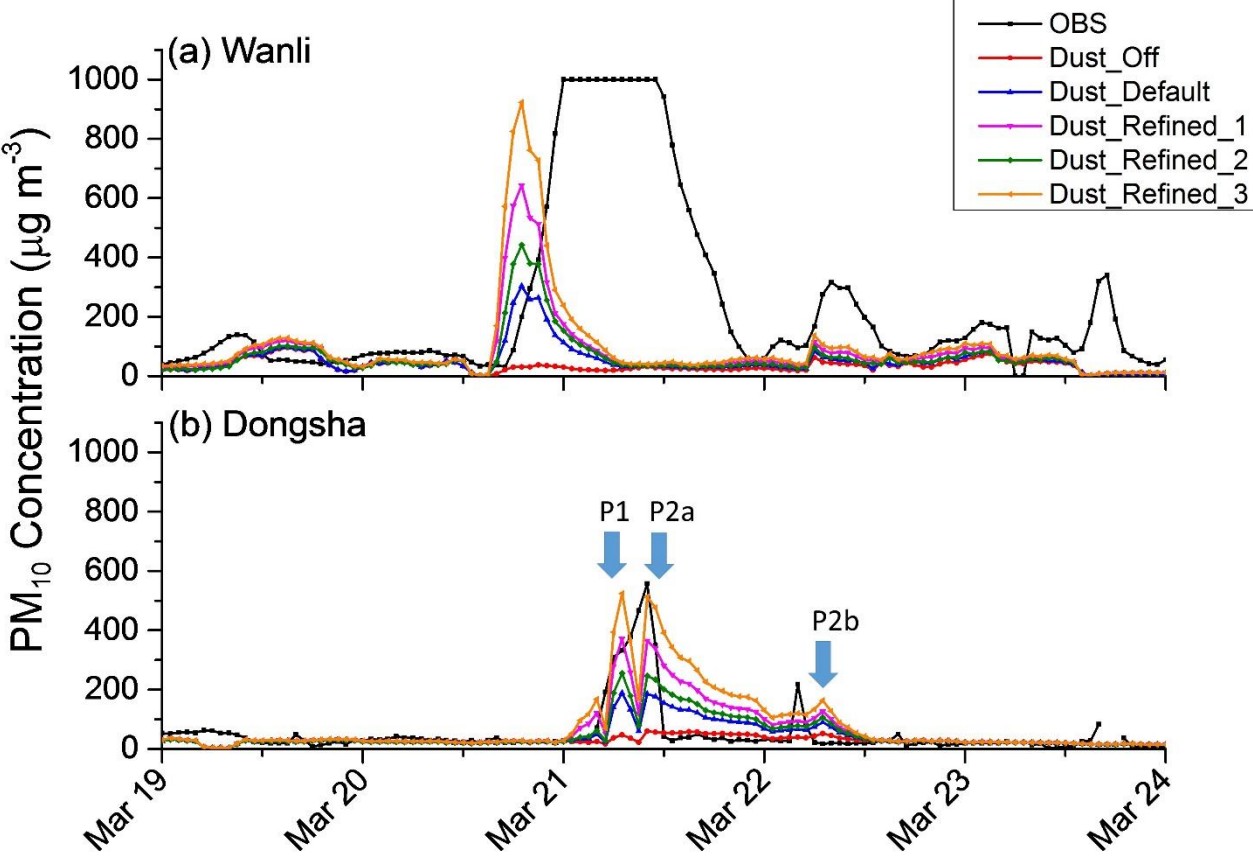

612
613
**Figure 2:** Time series of observed $PM_{10}$ concentrations over Wanli sites and Dongsha Island during 19-
23 March 2010.

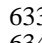


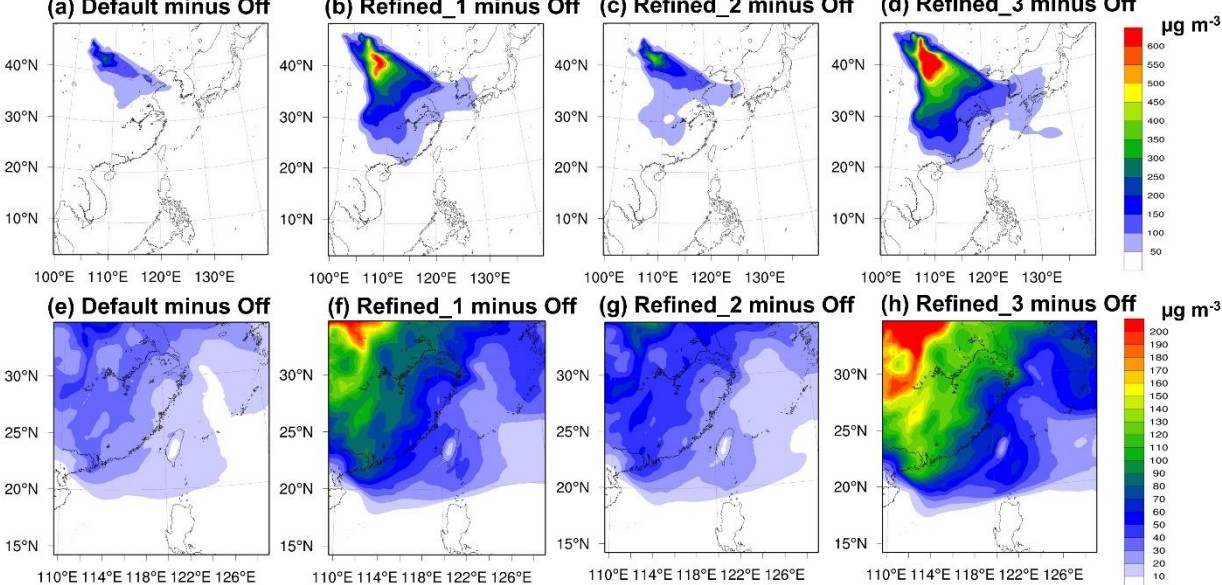

**Figure 3:** The difference of the daily average modeled PM$_{10}$ concentrations over d01 (a–d) and d02 (e–
h) between Dust_Off, and Dust_Default, Dust_Refined_1, Dust_Refined_2 and Dust_Refined_3,
respectively.

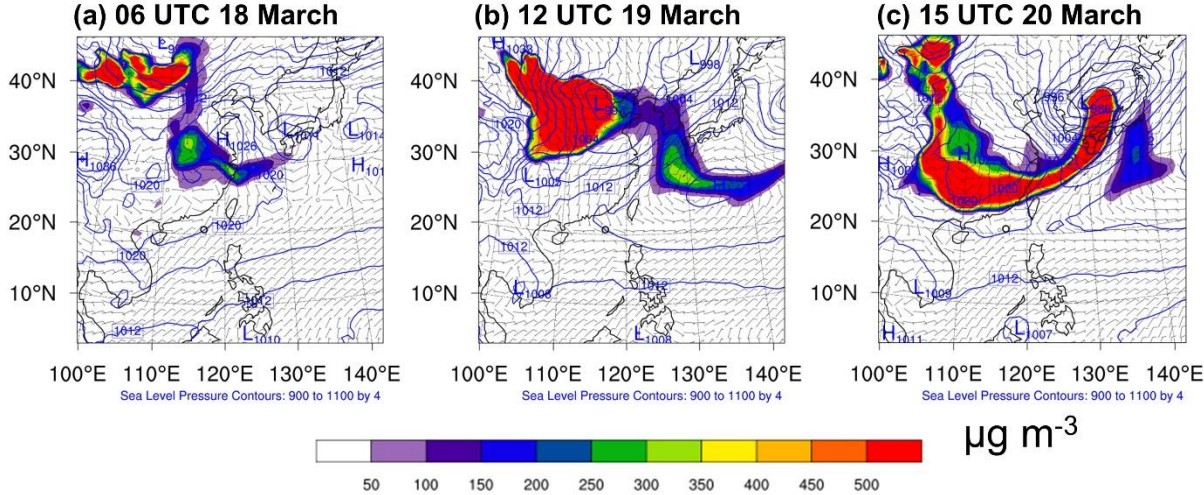

**Figure 4:** Spatial distribution of the simulated dust aerosol during (a) 06 UTC 18 March, (b) 12 UTC 19
March and (c) 15 UTC 20 March, in the year of 2010.




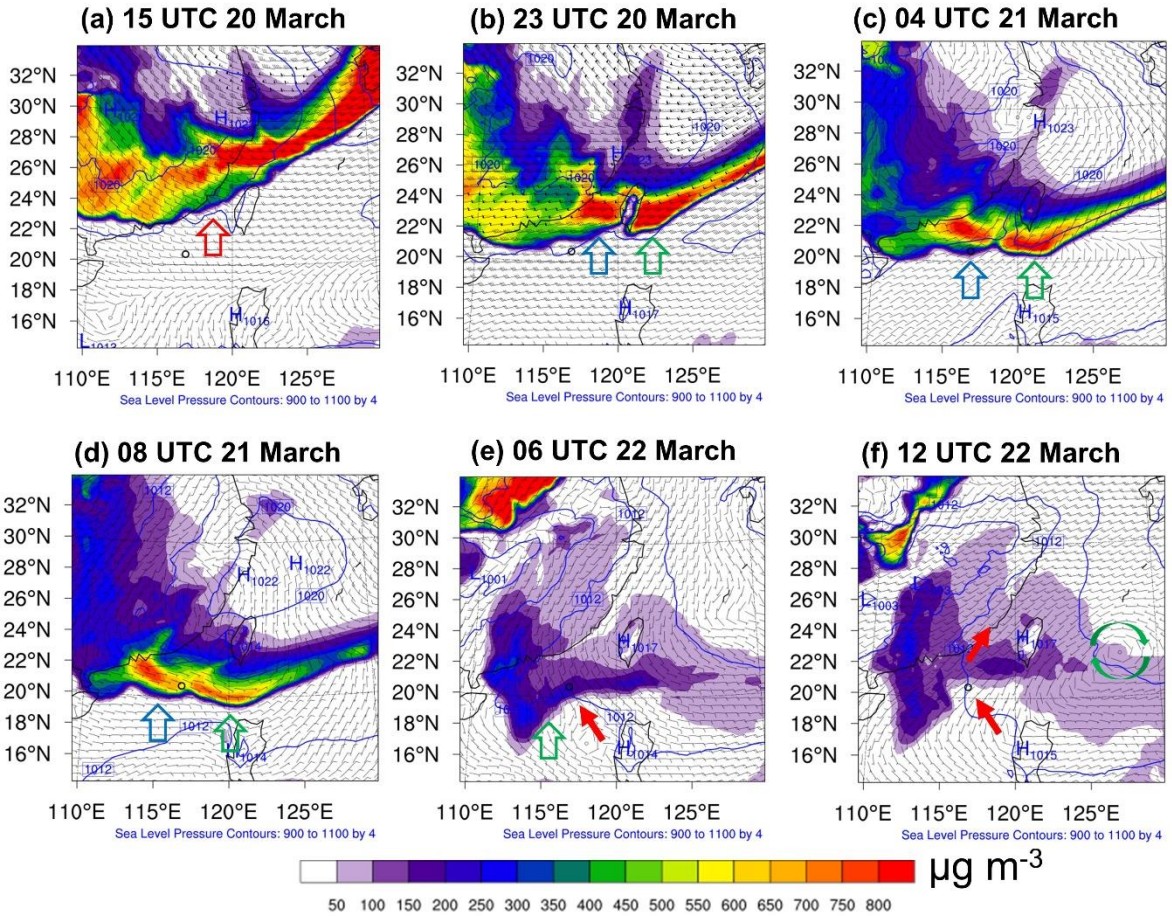

**Figure 5**: Spatial distribution of the simulated dust aerosol in the year 2010, during (a) 15 UTC 20
March, (b) 23 UTC 20 March, (c) 04 UTC 21 March, (d) 08 UTC 21 March, (e) 06 UTC 22 March and
(f) 12 UTC 22 March.




**Figure 6:** Vertical profile of dust aerosol for the CMAQ simulation of (a, c, e, g) control run and (b, d, f, h) without CMR at (a, b) 18 UTC 20 March, (c, d) 00 UTC 21 March, (e, f) 04 UTC 21 March and (g, h) 15 UTC 21 March in the year of 2010.



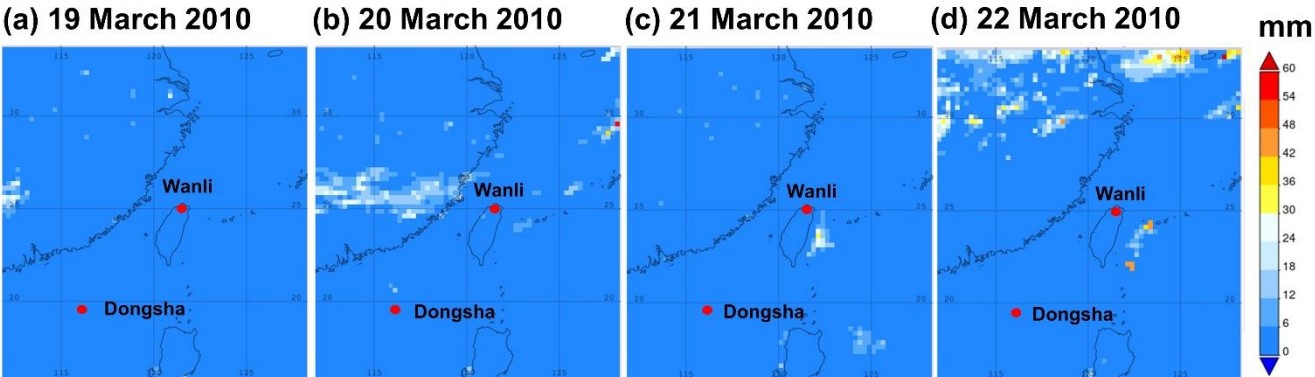

**Figure 7**: Daily mean merged precipitation data from the Global Precipitation Mission (GPM) satellite during 19-22 March 2010.

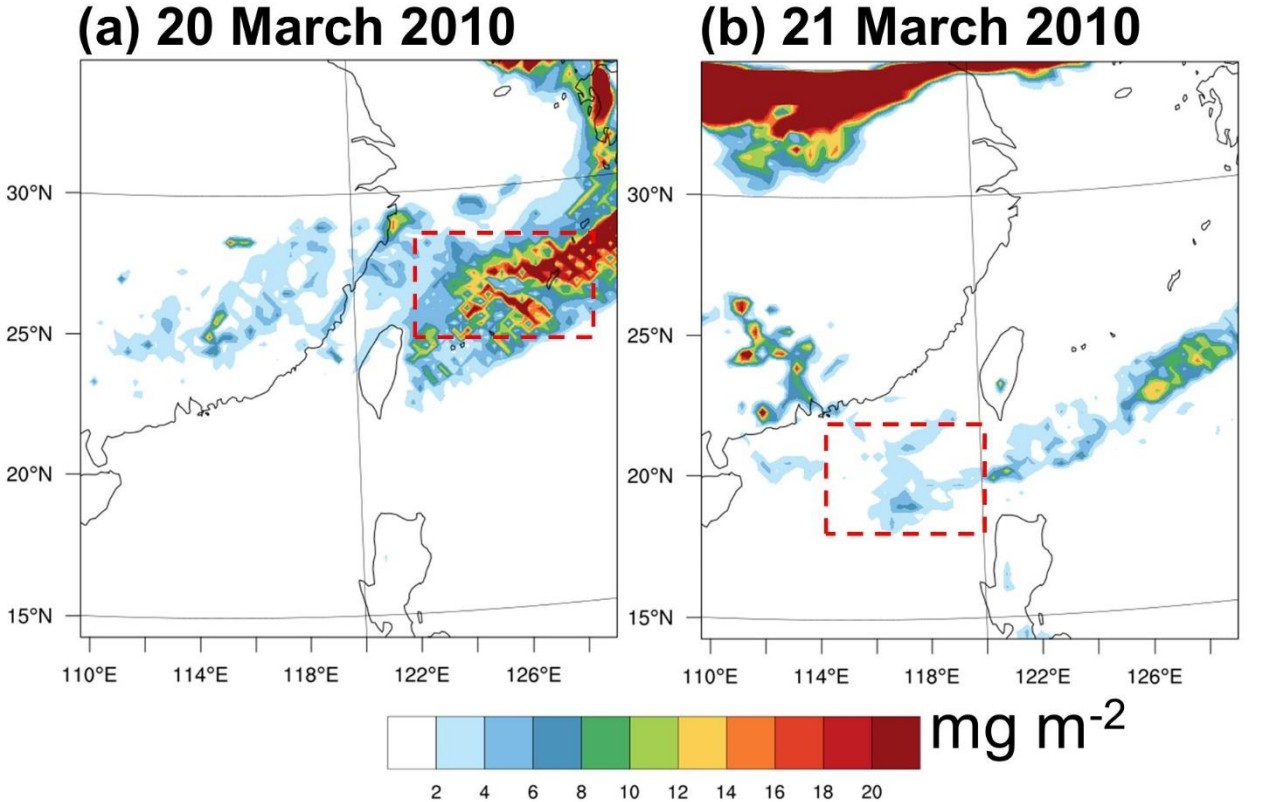

**Figure 8**: Spatial distribution of the wet deposition during (a) 20 March 2010 and (b) 21 March 2010.





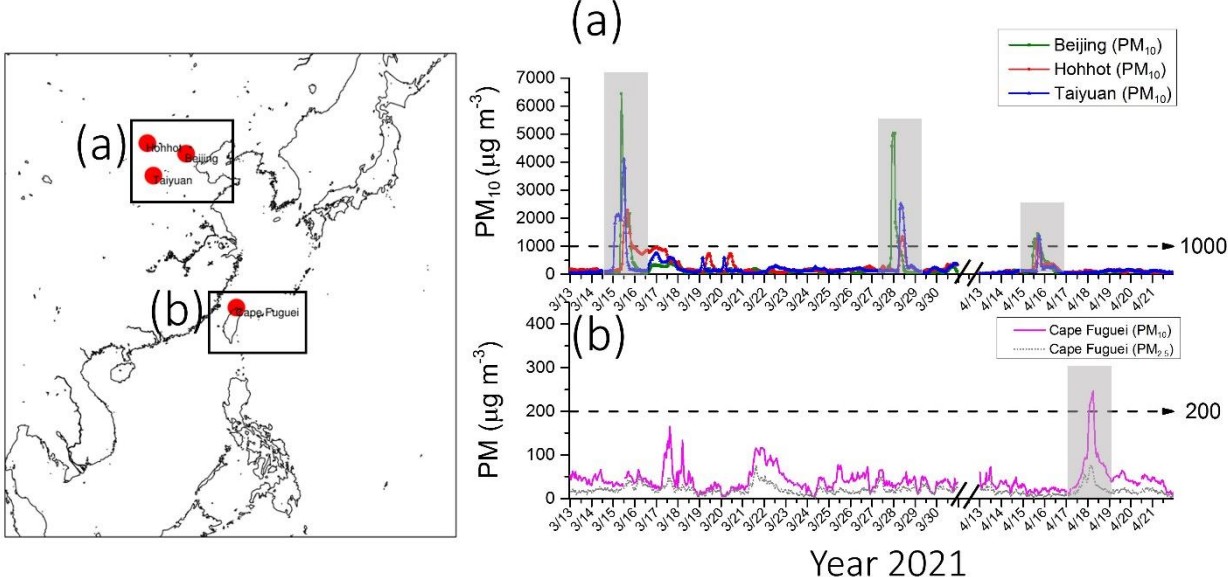

**Figure 9**: Time series of the observed PM$_{10}$ concentrations over the source region including (a) Beijing, Hohhot and Taiyuan; and the observed PM$_{10}$ and PM$_{2.5}$ at (b) Cape Fuguei during the spring 2021.



680

681

**Figure 10**: Spatial distribution of the MERRA-2 surface dust mass concentrations over the western North Pacific Ocean (shown in black rectangular box) during (a) 00 UTC 18 April, (b) 06 UTC 18 April, (c) 12 UTC 18 April, (d) 18 UTC 18 April, (e) 00 UTC 19 April, (f) 06 UTC 19 April, (g) 12 UTC 19 April and (h) 18 UTC 19 April 2021. The CMAQ surface dust mass concentrations during (i) 00 UTC 18 April, (j) 12 UTC 18 April, (k) 00 UTC 19 April and (l) 12 UTC 19 April 2021.




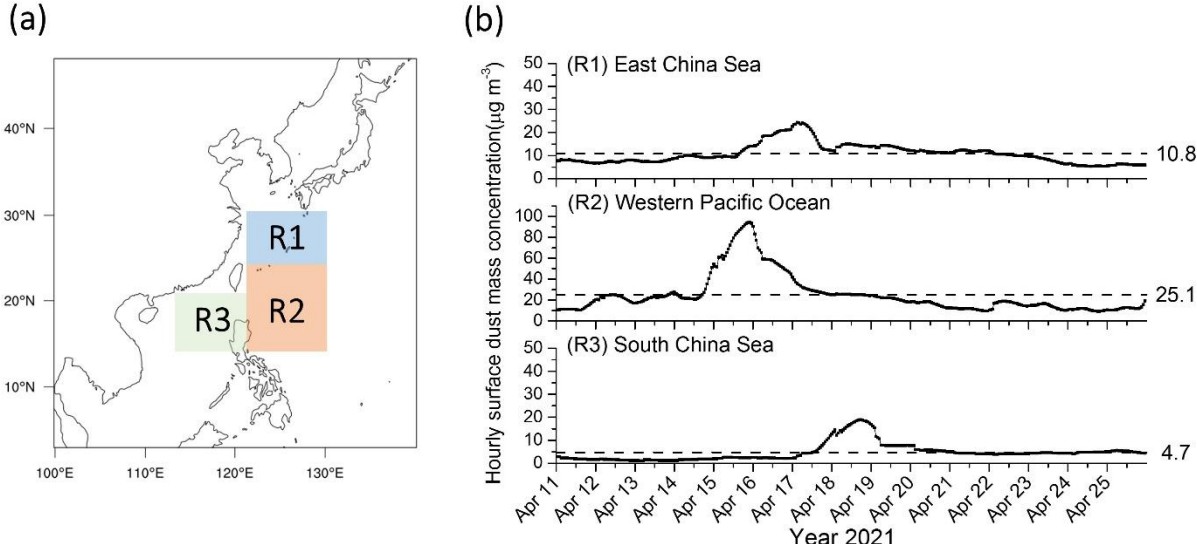

**Figure 11**: MERRA-2 hourly averaged dust mass concentrations over (a) R1: East China Sea, R2: Western Pacific Ocean and R3: South China Sea, during (b) 11-25 April 2021. Black dash line indicates the mean of dust mass concentration.

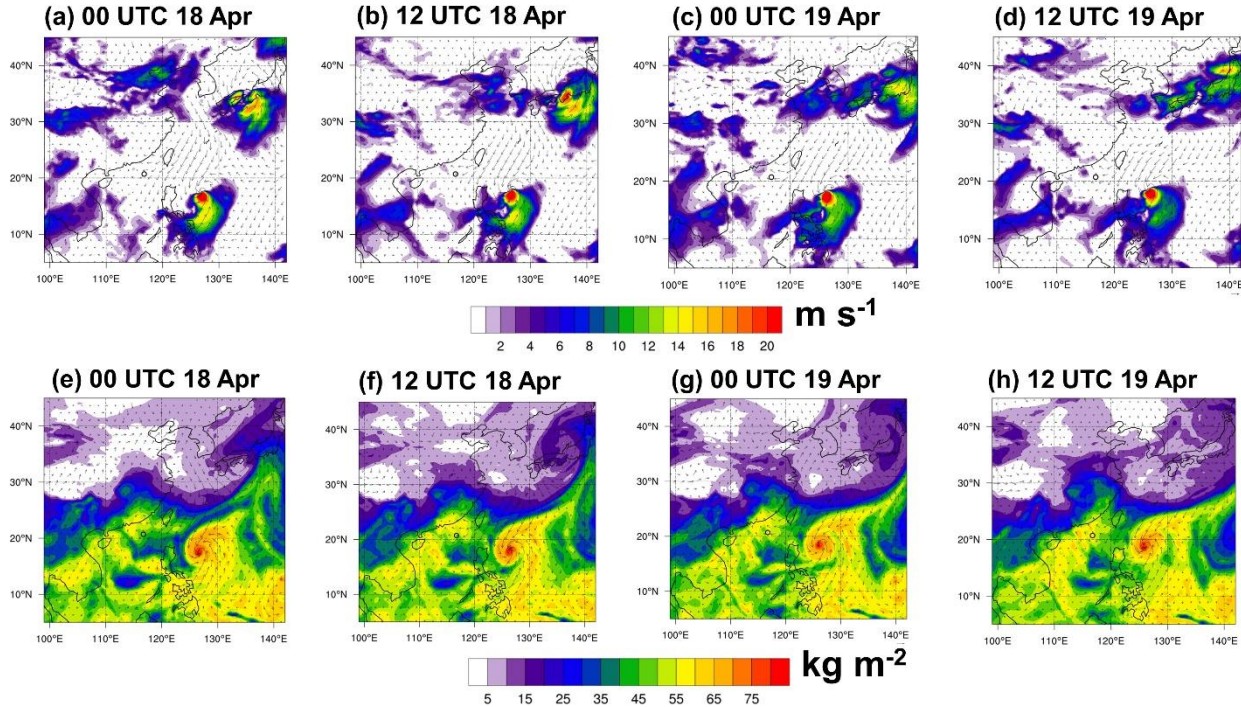

**Figure 12:** Spatial distribution of the MERRA-2 (a-d) wind speed and (e-h) total precipitation water vapour during (a, e) 00 UTC 18 April, (b, f) 12 UTC 19 April, (c, g) 00 UTC 19 April and (d, h) 12 UTC 19 April 2021.





**Figure 13:** Spatial distribution of the (a-d) wet deposition, (e-h) dry deposition, (i-l) average daily precipitation and (m-p) daily mean wind speed during (a, e, i, m) 18 April, (b, f, j, n) 19 April, (c, g, k, o) 20 April and (d, h, i, p) 21 April 2021.



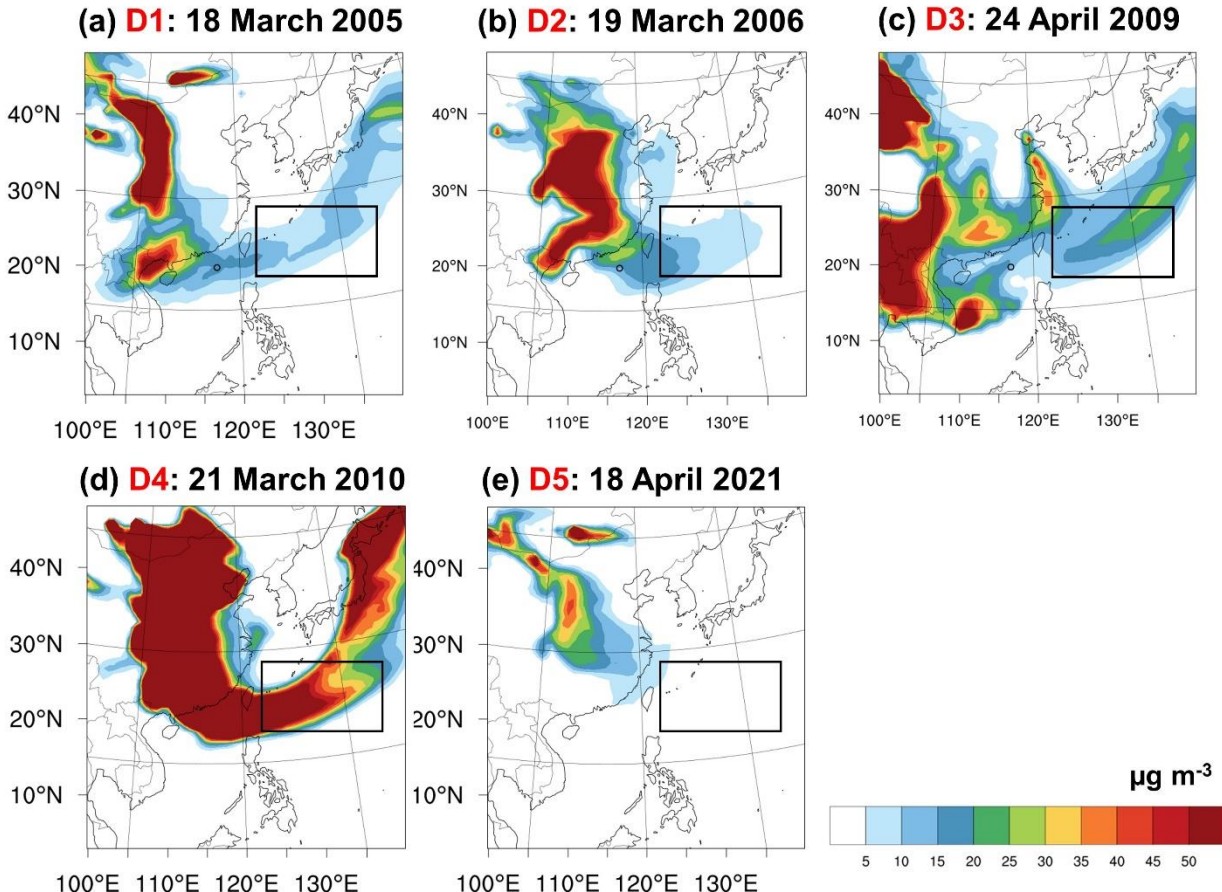

**Figure 14**: Daily mean surface dust mass concentrations for (a) D1: 18 March 2005, (b) D2: 19 March 2006, (c) D3: 24 April 2009, (d) D4: 21 March 2010 and (e) D5: 18 March 2021.