# Peer review of "Expanding the simulation of East Asian Super Dust Storm: Physical transport mechanism impacting the Western Pacific"

_EGUsphere, 2023_

## Author Comment (AC1)

**Reply to the comments of RC1**

**RC1: Overall comment**

Kong et al. present a refined model for simulating the atmospheric transport of dust. The authors new model is an amalgam of two previous iterations (Kong et al. 2021; Liu et al. 2021), finding that the combination of the previous results produces a superior model compared to either of the previous studies alone. Considering the noted improvements, and the possibility of increased extreme weather events in the future (a point that the authors may consider adding as an implication), the submitted manuscript should eventually be published in ACP. However, I do have some minor comments below that should be addressed first.

**Response:** The authors wish to thank the reviewer for his/her compliments, positive and constructive comments of our work. All of the changes in the revised manuscript have been highlighted in yellow. Corrections (blue text) with line numbers indicated in this response document refer to the revised manuscript.

Our point-by-point responses to the reviewer's comments are given below:

**Comment 1:** General: The figures need to be reordered, as the first figure referred to in the text is Figure 2 (Page 7 Line 172).

**Response:** We thank reviewer for the comment. Figure 1 is referring to the modelling domain, which we have mentioned in Page 4, line 97. Figure 2 shows the time series of measured and simulated $PM_{10}$ concentrations at the two measurement sites at Taiwan.

**Comment 2:** General: Figure captions generally lack sufficient detail and definitions.

**Response:** We have revised the captions of Figure 1, Figure 2, Figure 4, Figure 5, Figure 6 and Figure 7 with additional details.

**Comment 3:** General: Can the authors comment on potential interfering species to PM10 concentrations, and how that subsequently affects the resultant simulation (e.g., biological particles)?

**Response:** We thank the reviewer for the comment. During the long range dust transport in this study, mineral dust is the dominant component of the $PM_{10}$ concentration. In addition to natural dust, the strong wind could serve to mix in more anthropogenic aerosol pollutant (e.g. black carbon, $NO_x$ and $SO_2$) from China along the transport path towards the downwind region. However, contributions from anthropogenic pollutant were less compared to the dust aerosol concentration along this path particularly during the dust storm episode (Lin et al., 2012). In addition, previous studies showed dust and bioaerosol coexistence has occurred over the Western Pacific region (Iwasaka et al., 2009). About 10 % of the dust particle coated by microorganisms like pollen, bacteria, and plant and animal fragments on its surface, but the components are not embedded within the CMAQ model in this study, thus was assumed minor. We also believe that the marine air over East China Sea could influence the PM10 concentration over the downwind region such as Japan, Taiwan and South China Sea. For instance, the bioaerosols can be either dilute or deposit caused by the water vapour over the marine boundary layer. As a result, even if included in the simulation, the bioaerosol contribution to PM10 would likely be insignificant. As the present research mostly revealed

the special dust transport pattern and dynamics, the mixing process between dust, anthropogenic pollutants, and microorganisms has not been discussed. However, this would be a great research topic to explore going forward.

**References:**

Iwasaka, Y., Shi, GY., Yamada, M. et al. Mixture of Kosa (Asian dust) and bioaerosols detected in the atmosphere over the Kosa particles source regions with balloon-borne measurements: possibility of long-range transport, Air Quality, Atmosphere & Health 2, 29–38, https://doi.org/10.1007/s11869-009-0031-5, 2009.

Lin, C., Chou, C. C. K., Wang, Z., Lung, S., Lee, C., Yuan, C., Chen, W., Chang, S., Hsu, S., Chen, W., and Chen, S.: Impact of different transport mechanisms of Asian dust and anthropogenic pollutants to Taiwan, Atmospheric Environment, 60, 403–418, https://doi.org/10.1016/j.atmosenv.2012.06.049, 2012.

**Comment 4:** Page 7 Lines 172 – 182: I'm not convinced that the interpretation of the data from the Wanli station is appropriate given the signal saturation that is clearly observed for the PM10 concentrations. Even with the caveats that the authors provide, providing a numerical evaluation of their model (i.e., percent error) compared to the experimental Wanli data is highly misleading. At the very least, I recommend removing any numerical comparison with this result.

**Response:** Agree. Wanli station has been replaced by Shilin station to represent the northern Taiwan region, as Shilin station has a complete dataset during these episodes, was not signal saturated, and was still one of the most affected stations by the extreme event. At the Shilin station, the $PM_{10}$ concentrations were 60 µg m$^{-3}$ during 12 UTC 20 March and then peaked at 1517 µg m$^{-3}$ at 5 UTC the next day before decreasing to 60 µg m$^{-3}$ about 19 hours later.

We have replotted Figure 2, recalculated the statistical index in Table 2, and revised the text as follows: "DUST_Off and DUST_Default were similarly underestimated (Normalized Mean Bias (NMB) = -64.69 % and -54.09 %, respectively), compared with the observed values, which is consistent with the results of Dong et al. (2016) and Kong et al. (2021) that simulated moderate-intensity dust events. The Dust_Refined_1 and Dust_Refined_2 simulations exhibited improved accuracy (NMB = -41.18 % and -49.88 %, respectively), highlighting the importance of revising the dust treatment before simulating the SDS event over a downwind region (Kong et al., 2021). Moreover, the NMB for Refined_1 was lower than Refined_2 suggesting that simply calibrating the bulk soil density is not as effective as calibrating for soil moisture fraction and dust emission speciation. Eventually, Dust_Refined_3 resulted in the best performance (NMB = -30.65 %). Our results indicate the importance of including both calibration methods in order to reduce the model uncertainty.

Figure 2 shows the in-situ and CMAQ-simulated $PM_{10}$ concentrations at Shilin station (representing northern Taiwan) and Dongsha Island (representing the northern South China Sea region) during 19-24 March 2010. In both locations, the Dust_Off trend vastly underestimated the observations, whereas Dust_Default showed increased $PM_{10}$ concentrations but still resulted in an underestimation. The maximum $PM_{10}$ concentration at Shilin reached 1517 µg m$^3$. The CMAQ model predicted a peak $PM_{10}$ concentration of 1040.8 µg m$^3$, thus was 45.8 % lower than the observation result." **Page 7, Line 161-177**.

[Figure]

**Figure 2:** Time series of observed and simulated $PM_{10}$ concentrations over the Shilin site and Dongsha Island during 19-23 March 2010. P1, P2a and P2b show the peak values of the simulated $PM_{10}$ concentrations under the Dust_Refined_3 scenario.

**Table 2** Statistical index for $PM_{10}$ concentrations during 19-23 March 2010, for Taiwan Island (Shilin, Pinzhen, Hsinchu, Xitun, Xinying, Zhuoyin, Daliao) and Dongsha Island.

| | Benchmark | Off | Default | Refined_1 | Refined_2 | Refined_3 |
|---|---|---|---|---|---|---|
| MeanObs | | 178.80 | 178.80 | 178.80 | 178.80 | 178.80 |
| MeanMod | | 52.05 | 65.77 | 83.20 | 71.65 | 97.31 |
| NMSE | | 2.06 | 1.53 | 1.19 | 1.37 | 1.05 |
| MFB | ± 60% | -63.10 | -53.32 | -43.09 | -49.94 | -36.63 |
| NMB | ± 85% | -64.69 | -54.09 | -41.18 | -49.88 | -30.65 |
| NME | 85% | 64.69 | 60.10 | 57.28 | 58.94 | 55.16 |
| FAC2 | 0.5–2.0 | 0.71 | 0.84 | 0.99 | 0.88 | 1.12 |
| R | > 0.35 | 0.24 | 0.35 | 0.38 | 0.40 | 0.37 |

Note: the definition of the statistical formulas NMSE: Normalized Mean Square Error; MNB: Mean Normalized Bias; NMB: Normalized Mean Bias; NME: Normalized Mean Error; FAC2: Factor of Two; R: Correlation Coefficient.

**Comment 5:** Figure 3: While providing the differences between 'Dust_Off' and other various simulations makes for informative visualizations, it may still be helpful to show the absolute representations (i.e., separate simulations for Dust_Off, Dust_Default, Dust_Refined_1, Dust_Refined_2, and Dust_Refined_3) in the supporting information.

**Response:** The separate simulations have been included. We have modified the sentence as "Daily average modeled $PM_{10}$ concentration differences between Dust_Off and other

simulations over the East Asia region during 19-23 March 2010 are shown in Fig. 3, with the corresponding simulated absolute concentrations shown in Fig. S1." **Page 7-8, Line 182-184**.

[Figure]

**Figure S1**: Daily average modeled $PM_{10}$ concentrations over East Asia under the following simulation scenarios: (a, f) Dust_Off, (b, g) Dust_Default_1, (c, h) Dust_Refined_1, (d, i) Dust_Refined_2 and (e, j) Dust_Refined_3.

**Comment 6:** Page 8 Line 196 – 197: The authors have (perhaps unintentionally) opened up Pandora's box here: if the MERRA-2 reanalysis is subject to error due to a poorly captured wind component (something the authors postulate as a reason for differences in their simulation over East Asia), should it even then be used to validate their more positive results later on? I pose this as a bit of a rhetorical question. However, the very nature of events that the authors are interrogating surround the issue of wind, and if the method used to establish ground truth (MERRA-2) is subject to wind-related errors, then this should be interrogated further (i.e., to what extent is the use of MERRA-2 as a ground truth appropriate?).

**Response:** We thank the reviewer for the comment. The statement from the 2010 case was meant to highlight the importance of the wind dataset in depicting transboundary dust events over the region. In Section 3.4, we used MERRA-2 to study the impact of typhoon on dust over western Pacific. Since MERRA-2 captured the presence of the typhoon and the typhoon generated extreme wind speeds in the region, the dust pattern influenced by Typhoon was clearly depicted by MERRA-2. Please refer to Figure 11 for MERRA-2 wind speed during the typhoon.

The discussion of the simulated and MERRA-2 wind speed for the 2010 case has been rephrased. We modified the sentences as follows: "Generally, the model-simulated wind speeds were more than 2 m s$^{-1}$ greater than MERRA-2 wind speeds across much of East Asia during the SDS event in March 2010 (Fig. S3). Throughout the dust plume arrival to the SCS region, the simulated wind speeds were 8-12 m s$^{-1}$, while those from MERRA-2 were of much lower magnitude or nearly zero. As a result, the current study emphasizes the importance of the wind dataset to depict transboundary dust events over the region" **Page 8, Line 194-199**.

[Figure]

**Figure S3:** Simulated wind speed (upper panel) and MERRA-2 reanalysis wind speed (lower panel) during (a, e) 19 March, (b, f) 20 March, (c, g) 21 March and (d, h) 22 March 2010.

**Comment 7:** Page 8 Line 205: Change 'Dust_OFF' to 'Dust_Off'

**Response:** The phase has been changed. We have changed it as "The average AOD value of the DUST_Refined_3 yielded an NMB of -16.02 %, which was markedly better than DUST_Off (-26.09 %), DUST_Default (-25.24 %), DUST_Refined_1 (-19.58 %) and DUST_Refined_2 (-24.40 %)." **Page 8, Line 206-208**.

**Comment 8:** Figure 2: Define P1, P2a, and P2b

**Response:** The terms have been defined. We have changed Figure 2's legend as "**Figure 2:** Time series of observed and simulated $PM_{10}$ concentrations over the Shilin site and Dongsha Island during 19-23 March 2010. P1, P2a and P2b show the peak values of the simulated $PM_{10}$ concentrations under the Dust_Refined_3 scenario."

**Comment 9:** Figure 4: While the contours may be helpful, the authors more frequently refer to various land masses instead to describe the movement of the dust plume. Therefore, consider replacing the background graphic with one in which the visualization of land masses is more clear.

**Response:** The figure has been replotted with the new background graphic as follow:

[Figure]

**Figure 4:** Spatial distribution of the simulated dust aerosol during the March 2010 episode over East Asia within domain 1 (d01) at (a) 06 UTC 18 March, (b) 12 UTC 19 March and (c) 15 UTC 20 March; and domain 2 (d02) at (d) 15 UTC 20 March, (e) 23 UTC 20 March, (f) 04 UTC 21 March, (g) 08 UTC 21 March, (h) 06 UTC 22 March and (i) 12 UTC 22 March. Location of Dongsha is indicated with a black dot. The red arrows highlights the wind direction.

**Comment 10:** Figures 4 and 5: Consider combining these figures together, both for the sake of simplified visualization and simplified references within the text. For example, the prose on page 9 seems to present these figures as two distinct talking points despite the high degree of similarity in the provided information.

**Response:** The previous Figure 4 and 5 has been merged. Please see Figure 4 under the response to Comment #9.

**Comment 11:** Figure 5: There are various green and red arrows on the different tiles. The significance of these need to be described via a legend and within the figure caption.

**Response:** The red arrow has been described via a legend and within the figure caption. Please see Figure 4 under the response to Comment #9.

**Comment 12:** Figure 6: Both axes needs to be defined (i.e., need labels), and the x-axis readability needs to be improved.

**Response**: The axes of the figure (now Figure 5) have now been defined. We have replotted the figure as follow:

[Figure]

**Figure 5:** Vertical profile of dust aerosol for the CMAQ simulation of (a, c, e, g) control run and (b, d, f, h) without CMR at (a, b) 18 UTC 20 March, (c, d) 00 UTC 21 March, (e, f) 04 UTC 21 March and (g, h) 15 UTC 21 March 2010.

---

## Author Comment (AC2)

**Reply to the comments of RC2**

**RC2: General comment**

The authors improved the dust simulation over East Asia based on a few extreme dust events by refining soil moisture factor, dust emission speciation profiles, and bulk soil density. They also examined dust removal by a typhoon. The particulate matter from model simulations were evaluated against observations, which confirmed an improvement. Overall, the paper is well written but require some minor edits before publication in ACP

**Response:** The authors wish to thank the reviewer for his compliments, positive and constructive comments of our work. All of the changes in the revised manuscript have been highlighted in yellow. Corrections (blue text) with line numbers indicated in this response document refer to the revised manuscript.

Our point-by-point responses to the reviewer's comments are given below:

**Specific comment**

**Comment 1:** Line 349-352: While the study showed an improved NMB with refined dust simulations, the authors should explain the possible causes of the remaining biases and offer suggestions on methods that will further reduce the NMB in future studies. After all, the statistics without refinement still satisfy the benchmark.

**Response:** The suggested discussion has been added. We included the sentence: "As the improved NMB with the refined dust simulation still shows a degree of model underestimation, a calibration process to resolve the aerosol removal mechanism may be the most impactful in closing this gap." **Page 14, Line 353-355.**

**Comment 2:** Line 589 – 590: How are the thresholds in the benchmark defined? A general statistical benchmark may not be appropriate for dust model evaluations. Even the default simulations satisfy the benchmark, which makes the refined simulation in this study less significant.

**Response:** Agree. Since there is no regular statistical benchmark to justify the dust model evaluation, the present modeling study is referring to the threshold suggested by Emery (2001). We added the following sentence: "The threshold of the statistical index is based on Emery (2001)." **Page 7, Line 161.**

**Reference:**
Emery, C., Tai, E., and Yarwood, G.: Enhanced meteorological modeling and performance evaluation for two Texas ozone episodes. prepared for the Texas Natural Resource Conservation Commission, prepared by ENVIRON International Corp, Novato, CA, 2001

**Technical comments:**
**Comment 3:** Line 15: Do you mean "dust radiative flux"?

**Response:** To avoid confusion, we modified the term from "dust flux" to "dust transport".
**Page 1, Line 15.**

**Comment 4:** Line 22: It should be either "higher" or "lower in magnitude" since they are negative values

**Response:** Agree. The term has been rephrased as "The Dust_Refined_3 normalized mean bias of PM10 was -30.65 % for the 2010 SDS event, which was lower in magnitude compared to Dust_Refined_1 (-41.18 %) and Dust_Refined_2 (-49.88 %)." **Page 1, Line 21-23**.

**Comment 5:** Line 27: "On 15-21 April 2021" => "During 15-21 April 2021"

**Response:** The term has been modified as **"**During 15-21 April 2021, both CMAQ simulations and satellite data highlighted the influence of typhoon 'Surigae' on dust transport to downwind Taiwan and the Western Pacific Ocean (WPO)." **Page 1, Line 27-29.**

**Comment 6:** Line 28: "Typhoon" should be capitalized

**Response:** The term has been capitalized as **"**The CMAQ Dust_Refined_3 simulations further revealed a large fraction of dust aerosols were removed over WPO due to Typhoon 'Surigae'." **Page 2, Line 29-31.**

**Comment 7:** Line 586: "bulb" => "bulk"

**Response:** The term has been changed. **Page 23, Line 583**.

**Table 1** Summary of the design of the simulations used in the present study.

| Scenarios | Descriptions |
| --- | --- |
| Dust_Off | Without in-line calculation of dust. |
| Dust_Default | With the new default wind-blown dust treatment (Foroutan et al., 2017). |
| Dust_Refined_1 | Refined the soil moisture factor and the dust emission speciation profile for the Gobi Desert as suggested by Kong et al. (2021). |
| Dust_Refined_2 | Refined the bulk soil density according to China's soil type as suggested by Liu et al. (2021). |
| Dust_Refined_3 | Considering the both of Dust_Refined_1 and Dust_Refined_2. |

**Comment 8:** Line 603: "dot representing" => "dots represent"

**Response:** The term in the caption has been changed as "Figure 1: (a) Modeling domain configuration used in the present study. The red dots represent the location of the observation sites at Shilin and Dongsha. (b) The blue lines represent the transects that the dust plumes travelled along in this studies that are discussed in Section 3." **Page 24, Line 605**.

**Comment 9:** Line 645 - 646: The caption needs to mention PM10 concentrations. The same applied to Figure 5.

**Response:** We thank the reviewer for the comment. In Figure 4 (now merged with Figure 5 from the previous manuscript), instead of $PM_{10}$, the transboundary process is explained based on dust mass concentration in order to emphasize purely mineral dust from northwest China. The caption has been modified. We changed the caption as "Spatial distribution of the simulated dust aerosol during the March 2010 episode over East Asia within domain 1 (d01) at (a) 06 UTC 18 March, (b) 12 UTC 19 March and (c) 15 UTC 20 March; and domain 2 (d02)

at (d) 15 UTC 20 March, (e) 23 UTC 20 March, (f) 04 UTC 21 March, (g) 08 UTC 21 March, (h) 06 UTC 22 March and (i) 12 UTC 22 March. Location of Dongsha is indicated with a black dot. The red arrows highlights the wind direction." **Page 27, Line 639-643**.

**Comment 10:** Supplementary Figure S2: "3-days" => "three-day". Which figures are based on MODIS? Please be more specific.

**Response:** The caption of the supplementary figure has been modified as "Figure S4: The three-day mean averaged AOD over East Asia region, for CMAQ (a1-a5, b1-b5, c1-c5, d1-d5) and MODIS (a6, b6, c6, d6) during 17-19 March 2005 (a1-a6), 18-20 March 2009 (b1-b6), 25-27 April 2009 (c1-c6) and 20-22 March 2010 (d1-d6)."

---

## Author Response (AR2)

**Reply to the comments of reviewer**

*Reviewer's comment*

The authors made satisfactory revisions to the manuscript. After reviewing the figures, my only concern is the tiny texts of the city/town labels in Figure 8, which should be enlarged before it is accepted for publication.

**Response:** We thank the reviewer for complimenting our work. All of the changes in the revised manuscript have been highlighted in yellow. Corrections (blue text) indicated in this response document refer to the revised manuscript.

The texts of the city/town labels in Figure 8 has been enlarged as follow:

[Figure]

**Figure 8**: Time series of the observed $PM_{10}$ concentrations over the source region including (a) Beijing, Hohhot and Taiyuan; and the observed $PM_{10}$ and $PM_{2.5}$ at (b) Cape Fuguei during the spring 2021.